# Current Indications and Future Landscape of Bispecific Antibodies for the Treatment of Lung Cancer

**DOI:** 10.3390/ijms24129855

**Published:** 2023-06-07

**Authors:** Hugo Arasanz, Luisa Chocarro, Leticia Fernández-Rubio, Ester Blanco, Ana Bocanegra, Miriam Echaide, Ibone Labiano, Ana Elsa Huerta, Maria Alsina, Ruth Vera, David Escors, Grazyna Kochan

**Affiliations:** 1Medical Oncology Department, Hospital Universitario de Navarra, Instituto de Investigación Sanitaria de Navarra (IdiSNA), 31008 Pamplona, Spain; hugo.arasanz.esteban@navarra.es (H.A.);; 2Oncobiona Group, Navarrabiomed, Instituto de Investigación Sanitaria de Navarra (IdiSNA), 31008 Pamplona, Spain; 3Oncoimmunology Group, Navarrabiomed, Instituto de Investigación Sanitaria de Navarra (IdiSNA), 31008 Pamplona, Spain; 4Division of Gene Therapy and Regulation of Gene Expression, Centro de Investigación Médica Aplicada (CIMA), Instituto de Investigación Sanitaria de Navarra (IdiSNA), 31008 Pamplona, Spain

**Keywords:** lung cancer, bispecific, nanobodies, NSCLC, SCLC, targeted therapies

## Abstract

Bispecific antibodies are a promising type of therapy for the treatment of cancer due to their ability to simultaneously inhibit different proteins playing a role in cancer progression. The development in lung cancer has been singularly intense because of the increasingly vast knowledge of the underlying molecular routes, in particular, in oncogene-driven tumors. In this review, we present the current landscape of bispecific antibodies for the treatment of lung cancer and discuss potential scenarios where the role of these therapeutics might expand in the near future.

## 1. Introduction

Nowadays, numerous antibody-based proteins are being preclinically and clinically developed and have proven to be useful research, diagnosis, and therapy tools due to their particular properties, such as high specificity and affinity [1]. However, their large molecular weight (~150 kDa) and their challenging high-cost production limit their capacities. Thus, other novel strategies, such as nanobodies and bispecific antibodies, are being developed to overcome those limitations and improve their pharmacological properties and efficacy [2,3].

Classical antibodies or immunoglobulins are formed by two identical heavy and two identical light chains connected with disulfide bonds representing a Y-shaped molecule [4]. The heavy chain comprises four domains, and the light chain folds into two domains [5]. At the end of each chain is the antigen-binding fragment, which corresponds to the variable region of the antibody [1,4]. 

During the early 1990s, Hamers-Casterman and her team discovered a new type of antibody circulating in Camelidae (including camels and llamas) devoid of light chains that are called “heavy chain-only antibodies” [6]. Their heavy chain structure consists of two constant regions, a hinge region and the antigen-binding domain (VHH) [1]. The VHH is the structural and functional equivalent of the antigen-binding fragment of conventional antibodies [5]. It is also referred to as a nanobody or single-domain antibody and is considered to be the smallest antigen-binding unit of an antibody. Its small molecular size (~15 kDa) allows it to penetrate easily into tissues, cross the blood–brain barrier, and invade solid tumors [7,8]. In addition to their small size, other unique advantages, such as their remarkable stability against extreme temperatures, high pressure, chemical denaturants, low pH, or the presence of proteases, make nanobodies an attractive option over conventional antibodies [1,3,7,9]. Hence, nanobodies share characteristics of small molecule drugs and monoclonal antibodies, and they may be a promising alternative to classical antibodies in some applications [1]. Currently, many nanobody-based strategies are being developed for cancer, molecular imaging, infectious diseases, or inflammatory conditions, among other medical fields [3].

On the other hand, bispecific antibodies are molecules composed of one core unit and two binding units that are specific to two different epitopes, thus being able to attach to two targets simultaneously. The clinical applications of these antibodies are numerous, and they might be particularly useful in cancer because of the great complexity of this disease, with intertwined oncogenic signaling routes able to bypass single target inhibition upstream. Moreover, several clinical trials have demonstrated greater efficacy when patients receive combined targeted therapies, including CTLA4 plus PD-1-blocking antibodies or BRAF- and MEK-targeted antibodies, strongly supporting the potential benefit of this strategy [10,11,12,13,14,15]. 

Bispecific antibody development strategies can be bifurcated into two categories, the antigen x antigen type and the antigen x cell-engager type. Additionally, from the perspective of molecular format, bispecific antibodies can be classified into the “full antibody type” and the “BiTE type” (Figure 1). Depending on the molecular format, different development strategies should be required. For instance, the antigen x antigen bispecific type simultaneously targets two tumor-expressed antigens (TAAs), generally inhibiting two cancer signaling pathways to inhibit tumor growth. Of note, a particular subtype of bispecific antibodies has been named after the acronym BiTE (Bispecific T-cell engager). They are small molecules consisting of two fused scFvs without Fc region; one of them targets a (TAA), and the other one is specific to a T cell-surface receptor, generally CD3, one of the components of the T cell receptor (TCR). When a BiTE engages CD3 and the tumor-associated antigen, it induces T cell activation and proliferation while, at the same time, ensuring the immunological synapse [16] and enhancing T cell cytotoxicity for the recognition and elimination of tumor cells. Currently, several BiTEs are being developed for the treatment of cancer, the one targeting DLL3 and CD3 being the most promising one for the treatment of lung cancer, demonstrating enhanced T-cell cytotoxicity against DDL3+ tumor cells (NCT05882058).

## 2. Approvals in Oncology

In 2009, the first bispecific antibody, catumaxomab, was approved in the European Union as an intraperitoneal therapy for malignant ascites. Unfortunately, it was found to be toxic because of Fc-mediated off-target T-cell activation in the liver and was voluntarily withdrawn from the market. 

Further on, blinatumomab (Blincyto), a bispecific CD19-directed CD3 T-cell engager, received accelerated approval by the FDA in December 2014 and by the EMA in 2015 for the treatment of adult and pediatric patients with B-cell precursor acute lymphoblastic leukemia (ALL). Regular approval was obtained in July 2017 after the results of the phase III trial TOWER (NCT02013167), finding a benefit in overall survival (7.7 vs. 4.0 months, HR 0.71, *p* = 0.01) and event-free survival (6-months 31% vs. 12%, *p* < 0.001) compared with standard chemotherapy in patients with B-cell precursor ALL that had progressed to at least one line of therapy [17].

During the last few years, other bispecific antibodies have been approved for the treatment of hematological malignancies. BCMA-targeted therapies have proven to be effective in patients with multiple myeloma. Belantamab mafodotin (Blenrep) was the first one to receive authorization for its use in August 2020, shortly followed by teclistamab (Tecvayli), with a breakthrough designation in 2022 by the FDA and also approved by the EMA the same year based on the results of the phase Ib trial MajesTEC-1. Moreover, in patients with multiple myeloma, talquetamab, a bispecific antibody targeting GPRC5D and CD3, was named a breakthrough therapy after the results of phase I MonumenTAL-1.

For the treatment of relapsed follicular lymphoma, the anti-CD20/CD3 antibody mosunetuzumab (Lunsumio) was approved in June 2022 by the EMA as a therapy after progression to two previous lines of treatment. A single-arm phase II trial found a complete response rate of 60%, much higher than the 14% obtained with copanlisib in previous trials [18].

To conclude, bispecific antibodies have also proven effective for the treatment of solid tumors, and two drugs for the treatment of neoplasms other than lung cancer have been approved in 2022. Tebentafusp (Kimmtrak), targeting gp100, was approved by the FDA and EMA for the treatment of uveal melanoma with HLA-A*02:01. Phase III IMCgp100-202 compared tebentafusp with the therapy chosen by the investigator (pembrolizumab/ipilimumab/dacarbazine), and found a benefit in progression-free survival (31% vs. 19% at 6 months, *p* = 0.01) and overall survival (83% vs. 59% at 1 year). Lastly, cadonilimab (anti-PD1/CTLA4) was approved in China in June 2022 for patients with relapsed or metastatic cervical cancer after progression to platinum-based chemotherapy [19]. Even though a phase III trial is still ongoing, the approval was granted based on the promising results of a single-arm phase II with patients that had progressed to one or two lines of treatment, obtaining an overall response rate (ORR) of 33%, median progression free survival (mPFS) of 3.75 months and, more interestingly, median overall survival (mOS) of 17.51 months [20].

## 3. Bispecific Antibodies for the Treatment of Lung Cancer

Currently, 274 clinical trials are studying bispecific- or nanobody-based strategies for the treatment of solid tumors. Lung cancer patients meet the inclusion criteria in 158 of those clinical trials (Table 1), 5 in early phase I, 91 in phase I, 43 in phase I/II, 12 in phase II, 1 in phase I/II, 1 in phase III, and 4 is not applicable (Figure 2). A brief description of each bispecific antibody is available (Appendix A).

All of them are open-label (no masking), except for one phase II trial (NCT03501056), having quadruple masking (participant, care provider, investigator, and outcomes). Furthermore, many of those clinical trials specifically target lung cancers, such as NCT05116007 and NCT03319940, for small-cell lung cancer (SCLC) or NCT05360381 and NCT02609776 for non-small-cell lung cancer (NSCLC), for example. Many targets are being studied (Figure 2b).

In addition, a couple of diagnostic PET imaging clinical trials are being developed. NCT05156515 and NCT05436093 trials evaluate a non-invasive PD-L1 (APN09 drug) and CLDN18.2 (18F-FDG drug), targeting nanobodies labeled with PET radio-nuclide as a molecular imaging tracer for PET/CT scan, where 68Ga-THP-APN09 PET/CT and 18F-FDG PET/CT diagnostic approaches will be used to detect PD-L1 or CLDN18.2 tumor expression, with the aim of identifying patients who could benefit from anti-PD-L1 or anti-CLDN18.2 therapy. Both are being developed by Peking University Cancer Hospital and Institute.

### 3.1. Non-Small-Cell Lung Cancer

NSCLC is one of the tumor types with the higher incidence worldwide. It is usually classified into two groups, squamous and non-squamous. The greater knowledge of the mutational landscape that drives tumor progression in NSCLC allowed a further division based on gene mutation, particularly in non-squamous tumors. In this context, different targeted therapies have been developed, having demonstrated a higher efficacy with a more favorable toxicity profile compared with conventional treatments.

At this moment, amivantamab-vmjw is the only bispecific antibody available for the treatment of lung cancer. It is a human IgG1-based antibody that targets EGFR and MET, and it also induces Fc-dependent trogocytosis (an active transfer of a fraction of a cell to another, including the membrane and/or surface molecules) by macrophages and antibody-dependent cytotoxicity (ADCC) by natural killer (NK) cells [21]. It was granted accelerated approval by the Federal Drug Agency (FDA) in May 2021 and was approved by the European Medicines Agency (EMA) that same year for patients with NSCLC and *EGFR* ex20ins mutations that have progressed to platinum-based chemotherapy. Approval was granted on the basis of the results of the phase I CHRYSALIS, which included 81 patients, and reported an ORR of 40% and median duration of response (mDOR) of 11.1 months. The most frequent adverse events were rash (86%) and paronychia (45%), but no G3-4 toxicities surpassed 5%, hypokalemia (5%), rash (4%), PE (4%), diarrhea (4%) and neutropenia (4%) being the most common [22].

Some combinations of amivantamab with other drugs are also in the advanced stages of development. Phase II CHRYSALIS-2, evaluating the combination of amivantamab with the third generation EGFR TKI inhibitor lazertinib in patients with EGFR mutant NSCLC after progression to osimertinib and platinum-based chemotherapy, was presented in ASCO Congress 2022, describing an ORR of 33% with mDOR of 9.6 months, irrespective of the original mutation or the sequence of treatment. Toxicity was comparable with the one reported in the CHRYSALIS trial [23]. A confirmatory phase III trial called MARIPOSA-2 (NCT04988295), comparing chemotherapy plus amivantanab and Llzertinib with chemotherapy in patients that have progressed to osimertinib, is enrolling patients at this moment. Moreover, phase III trial MARIPOSA (NCT04487080), which will compare this combination with osimertinib or lazertinib monotherapy as a frontline treatment, is currently ongoing [24]. Finally, phase III trial PAPILLON (NCT04538664) is evaluating the benefit of the addition of amivantamab to platinum-based chemotherapy in patients with NSCLC and *EGFR* exon 20 insertions, and preliminary results are expected to be published within the next few years.

To conclude, zenocutuzumab, an HER2 and HER2 bispecific antibody, is also under evaluation, with a special focus on patients with NRG1 fusion. NRG1 is a membrane glycoprotein involved in cell growth and differentiation, which acts as a ligand for ERBB3 and ERBB4. Under common circumstances, NRG1 is cleaved by proteases and released in its mature form, limiting its activity. However, NRG1 fusions are poorly attached to proteases, favoring the accumulation of the protein in the membrane and its binding to HER3, causing heterodimerization with HER2 and downstream signal transduction. The combined results of the phase II part of the basket trial and the early expanded access program revealed an ORR of 34% among the 41 patients with NSCLC, with an mDOR of 9.1 months for the whole cohort, and less than 5% of G3 adverse events. 

### 3.2. Small-Cell Lung Cancer

SCLC is a neoplasm of neuroendocrine origin strongly associated with a smoking habit. It is characterized by a poor prognosis, with cancer cells presenting a very high proliferative rate and early metastatization. The high cellular heterogeneity, with a high mutation burden, is a major barrier to the incorporation of new treatments into the therapeutic arsenal, as targets expressed by all the cells are uncommon. From a molecular point of view, the inactivation of tumor suppressor genes defines this disease, with TP53 and RB1 being dysfunctional in most cases. 

In recent years, immunotherapy has been positioned in the frontline treatment of small-cell lung cancer, and at this very moment, two immune-checkpoint inhibitors, atezolizumab and durvalumab, are widely used combined with chemotherapy [25,26]. However, the efficacy is modest, probably due to the great cell plasticity and tumor heterogeneity, and new treatment strategies combining different approaches might be advantageous.

Delta-like ligand 3 (DLL3) is an inhibitory ligand of the NOTCH pathway frequently upregulated in SCLC that promotes cell invasion and metastases through epithelial-to-mesenchymal transition (EMT) [27]. Several early trials evaluating the efficacy of tarlatamab, a novel DLL3-targeted BiTE, in patients with SCLC, have reported appealing outcomes. In ASCO Annual Meeting 2021, the results of the phase I trial DeLLphi-300, in which patients with SCLC were treated with tarlatamab after progression to platinum-based chemotherapy, were presented. Even though 40% had previously received immune-checkpoint inhibitors (ICI) and 47% had liver metastases, usually associated with resistance to ICI, ORR was 20% with mDOR of 8.7 months, and disease control was achieved in 47%. The drug’s toxicity was manageable; 27% presented G3 treatment-related adverse events (TRAEs), which forced the interruption of the treatment in 7.6% of patients; additionally, 44% of patients experienced cytokine-release syndrome (CRS) [28]. The final results were recently published, reporting an ORR of 23.4% with an mDOR of 12.3 months, an mPFS of 3.7 months, and a mOS of 13.2 months. Tumor DLL3 expression was associated with better outcomes. The main TRAEs were CRS (52.3%), pyrexia (40.2%), constipation (30.8%), and 30.8% experienced toxicity ≥ G3 [29].

### 3.3. Toxicity of Bispecific Antibodies

Up to date, TRAEs caused by bispecific antibodies being evaluated for the treatment of lung cancer appear manageable, although the very structure of the BiTE tarlatamab and its immune-stimulating effect confers a less favorable toxicity profile compared with zenocutuzumab and amivantamab. 

Patients treated with tarlatamab experience G3-5 TRAEs more frequently than those receiving zenocutuzumab or amivantamab, with CRS and neurological adverse events (AEs) being particularly concerning. Moreover, it should be taken into account that most oncologists treating lung cancer might not have any previous experience in the management of either CRS or immune effector cell-associated neurotoxicity syndrome (ICANS), so an additional effort to ensure proper handling would be of great interest [29].

Regarding zenocutuzumab and amivantamab, besides the aforementioned TRAEs, infusion reactions were frequent, as high as 66% in patients receiving the latter [22]. Tight surveillance and patient education might be useful to adequately manage these episodes. 

## 4. Future Areas of Development

Some clinical scenarios might be of greater interest for the use of bispecific antibodies. Particularly in patients with mutation-driven NSCLC, both concurrent de novo gene alterations and their emergence as resistance mechanisms to targeted therapies are common. In patients with EGFR mutant NSCLC, concomitant MET, HER2, and PI3KCA alterations are the most frequent, reaching up to 19%, 5%, and 7% after progression to osimertinib, respectively. This is the rationale underlying the clinical trials with the EGFR/MET bispecific antibody amivantamab, but other drugs could also be evaluated in this context [30].

Regarding other oncogene-driven NSCLC, comparable resistance mechanisms can be found, although in different proportions. After progression to ALK-targeted therapies, common off-target mutations include EGFR alterations, HGF/MET pathway activation, and KIT amplification, among others [31]. Off-target mechanisms were detected in 45% of NSCLC patients with *MET* ex14 mutations after treatment with TKIs [32,33]. Alterations in different points of the RTK/RAS/MAPK/PI3K pathway were observed in 5 out of 10 patients with NSCLC and *KRAS* G12C mutation treated with adagrasib [34]. Moreover, RAS, MET, and BRAF were common targetable resistance mechanisms in patients with NSCLC and *RET* fusion treated with RET inhibitors [35,36].

Concerning immunotherapy treatment, a vast number of different immune checkpoints that play a role in the antitumor immune response suggest that combined blockade might derive greater efficacy. In fact, several trials evaluating the combination of drugs targeting different immune checkpoints in patients with NSCLC have been published. Anti-PD-1 plus anti-CTLA-4 schemes have reported better outcomes [26,37,38], although combinations, including anti-TIGIT or anti-LAG3, among others, are also being explored [39,40].

Of course, this also suggests that bispecific antibodies targeting several immune checkpoints could provide an advantageous strategy and improve efficacy in the treatment of cancer. MEDI5752 is a PD-1/CTLA-4 bispecific checkpoint inhibitor that has been evaluated in patients with different solid tumors. In ESMO Congress 2022 were reported the results of the phase I/II trial in patients with non-squamous NSCLC who received MEDI5752 combined with platinum and pemetrexed. The outcomes were promising, and in a small randomized cohort controlled with placebo, the efficacy was numerically higher in terms of ORR (50% vs. 47.6%, and the difference grew to 55.6% vs. 30% in PD-L1 negative tumors), mPFS (15.1 vs. 8.9 months), and mOS (non-reached vs. 16.5 months). However, due to high rates of toxicity, with 70% G3 adverse events, which led to discontinuation in 70%, a lower MEDI5752 dose of 750 mg instead of 1500 mg is under evaluation [41]. 

During the following years, the results of ongoing clinical trials are expected to confirm the efficacy of bispecific antibodies. A deeper knowledge of potential targets playing a capital role in tumorigenesis, immune system evasion, or resistance to treatment might entail a great impact on the development and usefulness of these promising therapies.

## 5. Conclusions

Even though the role of bispecific antibodies for the treatment of lung cancer has been limited to date, the deeper knowledge of the multiple genomic alterations driving tumor progression and treatment resistance suggests that the indications might experience a great expansion during the following years. The progression of mutation-driven tumors is caused by the persistent activation of a protein inducing uncontrolled cell proliferation and tumor growth. Current scientific guidelines recommend the screening of genetic alterations in several genes, including *ALK*, *BRAF V600*, *EGFR*, *HER2*, *KRAS G12C*, *MET*, *NTRK1-3*, *RET*, and *ROS1* in patients with NSCLC, as drugs targeting these proteins have demonstrated unprecedented efficacy. Concurrent mutations of other oncogenic genes are common, and the simultaneous blockade might be beneficial. Moreover, the treatment with tyrosine kinase inhibitors frequently induces off-target resistance mechanisms that require combined inhibition to prevent further tumor progression. With respect to immunotherapy, although the PD-1/PD-L1 axis has proven to be of utmost relevance, other immune checkpoints play major roles in the immune escape, as demonstrated by several clinical trials [42,43]). A single molecule able to inhibit several of these targets would reduce the complexity of patient management and follow-up. We expect Amivantamab-vmjw to be only the first in this group of drugs with the potential to significantly improve cancer treatment efficacy.

## Figures and Tables

**Figure 1 ijms-24-09855-f001:**
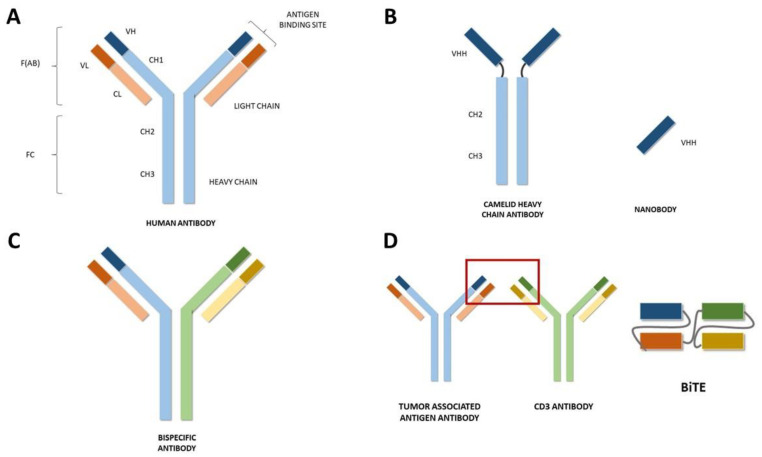
(**A**) Schematic representation of a conventional human antibody. Heavy chain is represented in blue, light chain is represented in orange. CH: Constant domain of heavy chain. CL: Constant domain of light chain. VH: Variable domain of heavy chain. VL: Variable domain of light chain. (**B**) Schematic representation of a nanobody. VHH: Single variable domain on a heavy chain. (**C**) Schematic representation of one modality of a bispecific antibody. The heavy and light chain specific for antigen 1 are represented in blue and orange, respectively. The heavy and light chain specific for antigen 2 are represented in green and yellow. (**D**) Schematic representation of a Bispecific T-cell Engager (BiTE).

**Figure 2 ijms-24-09855-f002:**
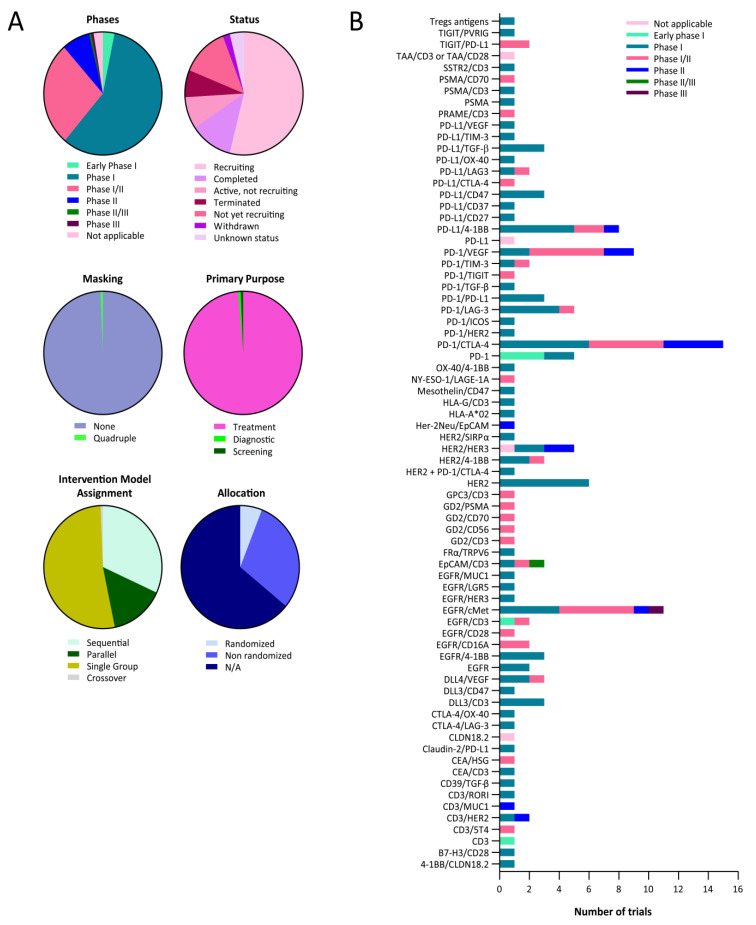
Clinical landscape of bispecific antibodies for the treatment of lung cancer. (**A**) Pie charts representing the proportions of clinical trials evaluating bispecific drugs for the treatment of lung cancer, according to the indicated categories. (**B**) Bar graph representing the number of clinical trials with bispecific antibodies for the treatment of lung cancer targeting the indicated antigens.

**Table 1 ijms-24-09855-t001:** Clinical trials evaluating bispecific antibodies in patients with lung cancer.

Target	NCT Number	Phase	Treatment
4-1BB/CLDN18.2	NCT04900818	Phase 1	TJ033721
B7-H3/CD28	NCT05585034	Phase 1	XmAbÂ^®^808
CD3	NCT04076137	Phase 1	T-cell armed with bispecific antibody
CD3/EGFR	NCT01081808	Phase 1	T cells armed with EGFR-bispecific
CD3/HER2	NCT02829372	Phase 1	CD3/HER2 bispecific antibody
CD3/HER2	NCT04501770	Phase 1	M802
CD3/RORI	NCT05607498	Phase 1	EMB07
CD39/TGF-β	NCT05381935	Phase 1	ES014
CEA/CD3	NCT02324257	Phase 1	RO6958688
Claudin-2/PD-L1	NCT04856150	Phase 1	Q-1802
CTLA-4/LAG-3	NCT03849469	Phase 1	XmAbÂ^®^22841
CTLA-4/OX-40	NCT03782467	Phase 1	ATOR-1015
DLL3/CD3	NCT03319940	Phase 1	AMG 757
DLL3/CD3	NCT05461287	Phase 1	QLS31904
DLL3/CD47	NCT05652686	Phase 1	PT217
DLL4/VEGF	NCT02298387	Phase 1	OMP-305B83
DLL4/VEGF	NCT03292783	Phase 1	NOV1501 (ABL001)
DLL3/CD3	NCT05361395	Phase 1	Tarlatamab
EGFR	NCT02687386	Phase 1	Mitoxantrone packaged EDV
EGFR	NCT02369198	Phase 1	TargomiRs
EGFR/4-1BB	NCT05442996	Phase 1	HLX35
EGFR/4-1BB	NCT05360381	Phase 1	HLX35
EGFR/4-1BB	NCT05150457	Phase 1	BNA035
EGFR/cMet	NCT04606381	Phase 1	Amivantamab
EGFR/cMet	NCT02609776	Phase 1	Amivantamab
EGFR/cMet	NCT04077463	Phase 1	Amivantamab
EGFR/CD3	NCT05387265	Phase 1	CX-904
EGFR/HER3	NCT04603287	Phase 1	SI-B001
EGFR/LGR5	NCT03526835	Phase 1	MCLA-158
EGFR/cMet	NCT02221882	Phase 1	LY3164530
EGFR/MUC1	NCT04695847	Phase 1	M1231
EpCAM/CD3	NCT04501744	Phase 1	M701
FRα/TRPV6	NCT04740398	Phase 1	CBP-1008
GPC3/CD3	NCT02748837	Phase 1	ERY974
HER2	NCT03842085	Phase 1	MBS301
HER2	NCT05320874	Phase 1	KM257
HER2	NCT04040699	Phase 1	KN026 and KN046
HER2	NCT03821233	Phase 1	ZW49
HER2	NCT02892123	Phase 1	Zanidatamab
HER2	NCT05380882	Phase 1	TQB2930
HER2 + PD-1/CTLA-4	NCT02760199	Phase 1	89Zr-AMG211
HER2/4-1BB	NCT03330561	Phase 1	PRS-343
HER2/4-1BB	NCT03650348	Phase 1	PRS-343
HER2/HER3	NCT00911898	Phase 1	MM-111
HER2/HER3	NCT01304784	Phase 1	MM-111
HER2/SIRPα	NCT05076591	Phase 1	IMM2902
HLA-A*02	NCT05359445	Phase 1	IMA401
HLA-G/CD3	NCT04991740	Phase 1	JNJ-78306358
Mesothelin/CD47	NCT05403554	Phase 1	NI-1801
OX-40/4-1BB	NCT04648202	Phase 1	FS120
PD-1	NCT05263180	Phase 1	EMB-09
PD-1	NCT05089266	Phase 1	CAR T cells
PD-1	NCT05373147	Phase 1	PD1-MSLN-CAR T cells
PD-1	NCT04503980	Phase 1	PD1-MSLN-CAR T cells
PD-1	NCT04489862	Phase 1	PD1-MSLN-CAR T cells
PD-1/CTLA-4	NCT04606472	Phase 1	SI-B003
PD-1/CTLA-4	NCT04572152	Phase 1	AK104
PD-1/CTLA-4	NCT05293496	Phase 1	Lorigerlimab
PD-1/CTLA-4	NCT03530397	Phase 1	MEDI5752
PD-1/CTLA-4	NCT03761017	Phase 1	Lorigerlimab
PD-1/CTLA-4	NCT03517488	Phase 1	XmAb20717
PD-1/HER2	NCT04162327	Phase 1	IBI315
PD-1/ICOS	NCT03752398	Phase 1	XmAb23104
PD-1/LAG-3	NCT04140500	Phase 1	RO7247669
PD-1/LAG-3	NCT03219268	Phase 1	Tebotelimab
PD-1/LAG-3	NCT05645276	Phase 1	AK129
PD-1/LAG-3	NCT05577182	Phase 1	INCA32459-101
PD-1/PD-L1	NCT03936959	Phase 1	LY3434172
PD-1/PD-L1	NCT04672928	Phase 1	IBI318
PD-1/PD-L1	NCT04777084	Phase 1	IBI318
PD-1/TGF-β	NCT05028556	Phase 1	Y101D
PD-1/TIM-3	NCT03708328	Phase 1	RO7121661
PD-1/TIM-3	NCT05357651	Phase 1	LB1410
PD-1/VEGF	NCT04047290	Phase 1	AK112
PD-1/VEGF	NCT05116007	Phase 1	AK112
PD-L1/4-1BB	NCT04009460	Phase 1	ES101
PD-L1/4-1BB	NCT04762641	Phase 1	ABL503
PD-L1/4-1BB	NCT03809624	Phase 1	INBRX-105
PD-L1/4-1BB	NCT04740424	Phase 1	FS222
PD-L1/4-1BB	NCT03922204	Phase 1	MCLA-145
PD-L1/CD27	NCT04440943	Phase 1	CDX-527
PD-L1/CD37	NCT04881045	Phase 1	PF-07257876
PD-L1/CD47	NCT04912466	Phase 1	IBI322
PD-L1/CD47	NCT04328831	Phase 1	IBI322
PD-L1/CD47	NCT05200013	Phase 1	BAT7104
PD-L1/LAG3	NCT05101109	Phase 1	BL501
PD-L1/OX-40	NCT05638334	Phase 1	S09501
PD-L1/TGF-β	NCT04958434	Phase 1	TST005
PD-L1/TGF-β	NCT05537051	Phase 1	PM8001
PD-L1/TGF-β	NCT04954456	Phase 1	QLS31901
PD-L1/TIM-3	NCT03752177	Phase 1	LY3415244
PD-L1/VEGF	NCT05650385	Phase 1	B1962
PSMA	NCT03927573	Phase 1	GEM3PSCA
SSTR2/CD3	NCT03411915	Phase 1	XmAb18087
TIGIT/PVRIG	NCT05607563	Phase 1	PM1009
Tregs antigens	NCT04156100	Phase 1	AGEN1223
CD3/5T4	NCT04424641	Phase 1/2	GEN1044
CD3/5T4	NCT05180474	Phase 1/2	GEN1047
CD3/GD2	NCT04750239	Phase 1/2	Nivatrotamab
CEA/HSG	NCT01221675	Phase 1/2	TF2
DLL4/VEGF	NCT04492033	Phase 1/2	CTX-009 (ABL001)
EGFR/CD16A	NCT05099549	Phase 1/2	AFM24
EGFR/CD16A	NCT04259450	Phase 1/2	AFM24
EGFR/CD28	NCT04626635	Phase 1/2	REGN7075
EGFR/CD3	NCT04844073	Phase 1/2	MVC-101 (TAK-186)
EGFR/cMet	NCT04868877	Phase 1/2	MCLA-129
EGFR/cMet	NCT04930432	Phase 1/2	MCLA-129
EGFR/cMET	NCT05498389	Phase 1/2	EMB-01
EGFR/cMET	NCT04590781	Phase 1/2	XmAb18087
EGFR/Cmet	NCT03797391	Phase 1/2	EMB-01
EpCAM/CD3	NCT05543330	Phase 1/2	M701
GD2/CD3	NCT03860207	Phase 1/2	3F8
GD2/CD56	NCT05437328	Phase 1/2	bi-4SCAR GD2/CD56 T cells
GD2/CD70	NCT05438368	Phase 1/2	bi-4SCAR GD2/CD70 T cells
GD2/PSMA	NCT05437315	Phase 1/2	bi-4SCAR GD2/PSMA T cells
HER2/4-1BB	NCT05523947	Phase 1/2	YH32367
NY-ESO-1/LAGE-1A	NCT03515551	Phase 1/2	IMCnyeso
PD-1/CTLA-4	NCT04172454	Phase 1/2	AK104
PD-1/CTLA-4	NCT03852251	Phase 1/2	AK104
PD-1/CTLA-4	NCT05559541	Phase 1/2	AK104
PD-1/CTLA-4	NCT05235542	Phase 1/2	AK104
PD-1/CTLA-4	NCT05505825	Phase 1/2	AK104
PD-1/LAG-3	NCT04618393	Phase 1/2	EMB-02
PD-1/TIGIT	NCT04995523	Phase 1/2	AZD2936
PD-1/TIM-3	NCT04931654	Phase 1/2	AZD7789
PD-1/VEGF	NCT04597541	Phase 1/2	Ivonescimab
PD-1/VEGF	NCT05689853	Phase 1/2	Ivonescimab
PD-1/VEGF	NCT05229497	Phase 1/2	Ivonescimab
PD-1/VEGF	NCT05214482	Phase 1/2	Ivonescimab
PD-1/VEGF	NCT04900363	Phase 1/2	Ivonescimab
PD-L1/4-1BB	NCT05159388	Phase 1/2	PRS-344/S095012
PD-L1/4-1BB	NCT04841538	Phase 1/2	ES101
PD-L1/CTLA-4	NCT05425602	Phase 1/2	MAX-40279-01
PD-L1/LAG-3	NCT03440437	Phase 1/2	FS118
PRAME/CD3	NCT04262466	Phase 1/2	IMC-F106C
PSMA/CD3	NCT04496674	Phase 1/2	CC-1
PSMA/CD70	NCT05437341	Phase 1/2	bi-4SCAR PSMA/CD70
TIGIT/PD-L1	NCT05102214	Phase 1/2	HLX301
TIGIT/PD-L1	NCT05390528	Phase 1/2	HLX301
CD3/MUC1	NCT03501056	Phase 2	CD3-MUC1 Bispecific Antibody
EGFR/Cmet	NCT05299125	Phase 2	Amivantamab
HER2/HER3	NCT02912949	Phase 2	Zenocutuzumab
HER2/HER3	NCT05588609	Phase 2	Zenocutuzumab
Her-2Neu/EpCAM	NCT00149019	Phase 2	Cell therapy
PD-1/CTLA-4	NCT04547101	Phase 2	AK104
PD-1/CTLA-4	NCT05377658	Phase 2	AK104
PD-1/CTLA-4	NCT05420220	Phase 2	KN046
PD-1/CTLA-4	NCT05215067	Phase 2	AK104
PD-1/VEGF	NCT04736823	Phase 2	AK112
PD-1/VEGF	NCT05247684	Phase 2	AK112
PD-L1/4-1BB	NCT05117242	Phase 2	GEN1046
EpCAM/CD3	NCT00836654	Phase 2/3	Catumaxomab
EGFR/CMET	NCT05388669	Phase 3	Amivantamab
CLDN18.2	NCT05436093	Not applicable	18F-FDG
HER2/HER3	NCT04100694	Not applicable	MCLA-128
PD-L1	NCT05156515	Not applicable	68Ga-THP-APN09
TAA/CD3 or CD28	NCT05119257	Not applicable	

## Data Availability

All the data used for this review is presented in the text.

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
