# Peer review of "Current Indications and Future Landscape of Bispecific Antibodies for the Treatment of Lung Cancer"

_ijms, 2023, doi:10.3390/ijms24129855_

Round 1
Reviewer 1 Report
The Authors presented the current landscape of bispecific antibodies for the treatment of lung cancer and discussed potential scenarios where the role of these therapeutics might expand in a near future. The review is original and interesting. However, I suggest to expand the discussion on the toxicity of these drugs that could be an issue in their application in clinical practice, adding a new paragraph (before paragraph 4) and reporting also the final data of the phase 1 study with tarlatamab (Paz-Ares L, JCO 2023). In this study, cytokine release syndrome occurred in
52% patients.
Author Response
We appreciate the reviewer’s suggestions. We included a paragraph adressing the potential toxicity of bispecific antibodies, and added the final results of the DeLLphi trial, which had not been published yet when we prepared the manuscript.
Reviewer 2 Report
The manuscript presents a review of bispecific antibodies in the context of lung cancer treatment. The content is commendable and potentially meets the acceptance criteria, although it currently appears to be more descriptive than analytical. The authors hint at the potential utility of this information for pharmaceutical companies, but are there any additional objectives for this work?
To enhance the clarity and comprehensiveness of this manuscript, there are several points that need to be addressed.
Bispecific antibody development is a hot topic in the field of oncology. The authors should more distinctly articulate how their review differs from others. For instance, a review article with similar content was published this year:
Chin Med J (Engl) 2023 Feb 20;136(4):379-393. doi: 10.1097/CM9.0000000000002460.
Bispecific antibody development strategies can be bifurcated into two categories: the antigen x antigen type and the antigen x cell-engager type. The authors need to elucidate the different criteria required for these two strategies.
Furthermore, from the perspective of molecular format, bispecific antibodies can be classified into the "full antibody-type" and the "BiTE type." Depending on the molecular format, different development strategies should be required. This should be clarified in greater detail by the authors.
Overall, I do not aim to critique the authors unnecessarily, but the manuscript seems to be a compilation of information from Beacon and other databases. It would be valuable to gain insight into the authors' unique perspectives.
Author Response
We are thankful for the reviewer’s comments.
- We understand the reviewer point of view. However, as that topic is very extensive and complex, we believe that our review could offers a complementary approach to the current bibliography on a cutting-edge, growing hot topic as it is the current indications and future landscape of bispecific antibodies for the treatment of lung cancer, and that if will serve as a useful bibliography tool to researchers and clinicians interested in learning more about this expanding field.
- The classification of antigen x antigen type and the antigen x cell-engager type has been added. A figure describing the differences between different categories has been added (Figure 1). Same for the “full antibody-type” and the “BiTE type”, stating the differences required for each case. The information presented about BiTEs antibodies has been extended. A figure describing the differences between different categories has been added (Figure 1).
Reviewer 3 Report
This review summarizes the current knowledge of using immunotherapy in lung cancer, focusing on bispecific antibodies. The manuscript is well written, it is easy to read, and to understand.
Comments:
(1) In the Introduction. Could you please show in a diagram the structure of a classical antibody, a nanobody (single domain antibody), and bispecific antibodies?
(2) Line 59. When you write "anti-CTLA4", do you mean that the antibody is specific and attaches CTLA4 molecule? But is it agonistic, or antagonistic? Some additional definitions may be useful to readers.
For example, this webpage is quite clear:
https://www.creative-biolabs.com/agonistic-antibody-therapy.html
(3) The diagram ob BiTE bispecific antibodies would be useful as well.
https://en.wikipedia.org/wiki/Bi-specific_T-cell_engager
(4) In Table 1, could you please add the function of the treatment/antibody that is being used?
For example:
TJ033721 is a Tetravalent IgG(H)-scFv fusion-type of bi-specific antibody (BsAb).
XmAb®808 is a Monoclonal bispecific antibody. XmAb808 is a tumor-selective, co-stimulatory XmAb 2+1 bispecific antibody designed to bind to the broadly expressed tumor antigen B7-H3 and selectively to the CD28 T-cell co-receptor (Signal 2) only when bound to tumor cells, which was demonstrated in in vitro studies. In vivo studies further demonstrated strong potentiation of checkpoint and CD3 cytotoxic activity.
NTC04076137, bispecific antibody armed anti-CD3-Actibated T cells(ATC).
(4) Line 135. Could you please define trogocytosis?
(5) In section 3.1. Could you please provide a background information of the clinicopathological characteristics of NSCLC?
(6) In section 3.2. Could you please provide a background information of the clinicopathological characteristics of SCLC?
(7) The manuscript would benefit from an explanation of the immune checkpoint pathways, molecules, and functions of the host immune response.
(8) Line 233. Could you please expand the information of the pathological mechanism in cancer of "mutation-driven tumors"? Currently, companion diagnostic in lung cancer include the screening of several mutations.
(9) Line 237. Could you please show a survival plot showing the different overall or progression free survival of lung cancer patients based on the PD1/PDL1 immunotherapy?
Author Response
We are thankful for the reviewer’s comments, and we hope the paper has improved enough to be suitable for publication.
1. A figure has been included.
2. Corrected.
3. A figure has been included.
4. We have included the information in a supplementary figure.
5. A definition has been included.
5. A brief description has been included.
6. A brief description has been included.
7. We agree with the reviewer. However it is not possible for us to include that information due to space limitations, as that topic is very extense and complex.
8. We have expanded the information regarding mutation-driven tumors.
9 Unfortunately we do not have the individual data of the patients participating in the main randomized trials evaluating the efficacy of PD-1/PD-L1 immunotherapy to generate the survival curves. However our review is mostly focused on bispecific antibodies and, although presenting that information would be of interest to the reader, we believe it is not capital for our work.
Round 2
Reviewer 2 Report
Accept in present form